# Nanoquercetin and Extracellular Vesicles as Potential Anticancer Therapeutics in Hepatocellular Carcinoma

**DOI:** 10.3390/cells13070638

**Published:** 2024-04-05

**Authors:** Alok Raghav, Goo Bo Jeong

**Affiliations:** Department of Anatomy and Cell Biology, College of Medicine, Gachon University, 155 Getbeol-ro, Yeonsu-gu, Incheon 21999, Republic of Korea; alokalig@gachon.ac.kr

**Keywords:** quercetin nanoparticles, extracellular vesicles, hepatocellular carcinoma, therapeutic targets, inhibitors, delivery

## Abstract

Despite world-class sophisticated technologies, robotics, artificial intelligence, and machine learning approaches, cancer-associated mortalities and morbidities have shown continuous increments posing a healthcare burden. Drug-based interventions were associated with systemic toxicities and several limitations. Natural bioactive compounds derived nanoformulations, especially nanoquercetin (nQ), are alternative options to overcome drug-associated limitations. Moreover, the EVs-based cargo targeted delivery of nQ can have enormous potential in treating hepatocellular carcinoma (HCC). EVs-based nQ delivery synergistically regulates and dysregulates several pathways, including NF-κB, p53, JAK/STAT, MAPK, Wnt/β-catenin, and PI3K/AKT, along with PBX3/ERK1/2/CDK2, and miRNAs intonation. Furthermore, discoveries on possible checkpoints of anticancer signaling pathways were studied, which might lead to the development of modified EVs infused with nQ for the development of innovative treatments for HCC. In this work, we abridged the control of such signaling systems using a synergetic strategy with EVs and nQ. The governing roles of extracellular vesicles controlling the expression of miRNAs were investigated, particularly in relation to HCC.

## 1. Background

Liver cancer ranks sixth as one of the most prevalent worldwide cancers, and 90% of liver cancers arise from HCC [1,2]. Hepatitis B and C, obesity, and excessive alcohol consumption are known risk factors for the progression of HCC [3]. Among these risks, hepatitis B alone is an important risk factor for cancer, accounting for nearly half of the cases [4,5]. Moreover, western countries showed an increased trend of non-alcoholic steatohepatitis (NASH) caused by metabolic diseases, adding to the risk of HCC development [6,7]. It was observed that older patients presented a greater number of NAFLD-related HCC than virus-concomitant HCC [8]. Patients with time-associated gut microbe population modification who have NAFLD are also thought to have a significant risk of developing HCC [9].

Liver cancer is one of the primary cancers with a high death rate, as stated by the World Health Organization (WHO), with an expected mortality of 0.83 million by the year 2020 [10]. HCC accounted for half a million mortalities and nearly more than half a million new cases in 2020 in Asia [11,12], with an increased trend in males compared to females. Furthermore, liver cancer had the seventh-highest incidence and sixth-highest deaths by the year 2020 in Asian women [10]. The average annual change (AAPC) in South Korea (−2.2%), Japan, China (−1.6%), and the Philippines (−1.7%) showed a decrease in liver cancer compared to other types [12]. Another GLOBOCAN 2020 assessment found Iran, Afghanistan, Qatar, Azerbaijan, Iraq, and Nepal to be at high risk [13]. Morbidity and mortality are higher in East Asia, North Africa, and Micronesia [14,15] (Figure 1, Figure 2 and Figure 3). As the episodes of HCC are continuously increasing, it is time to develop and focus on new technologies.

Cancer of the Liver Italian Program (CLIP), Hong Kong Liver Cancer (HKLC), Okuda, Barcelona Clinic Liver Cancer (BCLC) (Figure 4), and the American Association for the Study of Liver Diseases (AASLD) are among HCC definitions and staging models [16,17]. Other classification regimes entail classifying HCC based on molecular genetics, metabolism, immunological characteristics, and chromosomal configurations [18].

Drugs like sorafenib, lenvatinib, atezolizumab, bevacizumab, and doxorubicin (DOX) have many limitations [19]. Such unsatisfactory results in the management of HCC can be linked to a variety of factors, including drug-related side effects, limited bioavailability, significantly raised toxic nature, untargeted transport of therapeutic agents, high expenditure of large-scale manufacture, immunological problems, and anaphylactic reactions [20]. To astound such precincts, researchers are now looking to explore possibilities of using nanoparticles and extracellular vesicles (EVs) as drug conveyance vessels to treat cancers. The present review will focus on exploring the potential of EVs and quercetin nanoparticles (Qnps) to treat HCC. Moreover, the pathways inhibiting the pathogenesis of HCC with Qnps and EVs are also explored. We inspected the substantiation auxiliary for the consumption of biological molecules, including EVs and quercetin, for treatment regimes for HCC. Finally, we performed a critical analysis of current clinically approved drugs/immunotherapies for HCC and ongoing preclinical/clinical trials using EVs and Qnps as anticancer agents in patients with HCC. 

## 2. Pathophysiology of HCC

HCC pathophysiology reveals multifarious multifactorial contrivances. The development of malignant transformation of hepatocytes depends on the interaction of many elements, including genetic influences, infectious and non-infectious diseases, and the rigorousness of liver disease along with the early stages of the cellular microenvironment. Studies have shown that approximately 80% of patients with cirrhosis have HCC due to molecular variations [2]. Viral factors include etiological infections linked with C and B hepatitis virus, whereas non-viral aspects include liquor intake, NASH, aflatoxin usage, tobacco, and aristolochic acid, which is known as a potent carcinogenic trigger [21]. Moreover, certain particular immunological and molecular reasons have been identified as initiators of HCC [21]. In this regard, research into checkpoints was required to understand the onset, progression, and therapy with biopharmaceuticals. However, major milestones have been thoroughly researched elsewhere [2,22,23].

## 3. Molecular Triggers of Hepatocellular Carcinoma

In liver cirrhotic patients, neoplasm advancement occurs via a consecutive flow of histopathological intonations, thereby instigating HCC progression. HCC-associated histomorphological characteristics comprise extremely vascularized tumors with conspicuous acinar and wide trabeculae along with loss of Kupffer cells and reticulin network [24]. In the HCC advanced stage, the tumor appears to be captured by CD34 and α-smooth muscle actin (SMA)-positive septa. Liver stromal cells, as well as mature hepatocytes, are the main cells in the origin and expansion of HCC [23,24,25,26]. A study testified that recurrent stress on regenerated hepatocytes causes genetic lesions, which induces transformation and oncogenesis development [27]. A previous study showed the role of cyclin-A2 or E1 proteins in HCC development, particularly in non-cirrhotic participants, furthermore mediated by activation of the E2F and ATR factors, as well as inactivation of RB1 and PTEN [27]. 

CD8+PD1+ T lymphocytes in NASH patients enhance hepatocyte death, enhancing the HCC microenvironment formation and advancement [28]. On the contrary, genomic, somatic, and epigenetic variations all contribute to HCC progression. A study testified that single nucleotide polymorphisms (SNPs) of PNPLA3 (rs738409), TM6SF2 (rs585542926), and HSD17B13 (rs72613567) contribute to predisposition to HCC [29]. Aflatoxin B1 and aristolochic acid (which promotes T to A inversion) have been shown to cause somatic mutations, increasing the likelihood of HCC development [30].

## 4. Checkpoint Targets of HCC

HCC progression results from a number of mechanisms that have many diagnostic sites and can be investigated as treatments for HCC. The following clinical sites are considered to have a role in HCC.

### 4.1. Wnt–β-Catenin Signaling

#### 4.1.1. CTNNB1

*A CTNNB1*-allied dynamic mutation is a key component of Wnt signaling and is unveiled in nearly 11–41% of liver cancer patients [31,32,33,34]. *CTNNB1* is dynamically tangled in the production of the actin cytoskeleton, which is liable for stumbling cell division [35]. Mutations in CTNNB1 have been recounted to be associated with the TERT promoter region, NFE2L2, MLL2, ARID2, and APOB [36,37]. Studies of human HCC have shown that mutations in CTNNB1 occur simultaneously with the modulation of Met, Myc, or Nrf2 genes [38,39,40]. It has been found that sorafenib and gamma-secretase inhibitors act as effective targets for exploiting the regulation of CTNNB1 [30,41]. 

#### 4.1.2. Adenomatous Polyposis Coli (APC) 

Mutations in Adenomatous Polyposis Coli (APC) ensue in the dominant section of the open reading frame (ORF), also recognized as the mutation cluster region (MCR), resulting in the shortening of the protein [42]. This process results in the deletion of numerous elements, comprising the β-catenin binding domain (20R), the nuclear localization sequence (NLS), the axin binding domain (ABS), and the C-terminal basic domain (CTBD), which is important for cytoskeletal interfaces. Mutations of sporadic APC are thought to cause cancer. Mutations in APC alter the Wnt-β-catenin signaling pathway, leading to the development and progression of HCC. 

#### 4.1.3. AXIN1

AXIN1 mutations have been shown to be associated with approximately 5–19% of advanced lung cancers [32,33]. AXIN1 inhibits Wnt/β-catenin signaling by modulating β-catenin expression [43]. One study found that overexpression of AXIN1 (wild-type) inhibited cell propagation in HCC while inducing programmed cell death, thus making it a molecular target for HCC therapy [43]. In another study, the authors employed adenovirus-mediated gene transfer of AXIN1 to induce apoptosis in HCC cells [44]. AXIN was discovered to inhibit tankyrases 1 and 2 via XAV 939, making it a potential therapeutic target in Wnt signaling [45]. 

### 4.2. Telomere Maintenance

#### TERT

Mutations of the TERT promoter were recognized to be allied with HCC pathogenesis. Studies have testified the mutations of the TERT promoter at −124 (G > A) and −146 (G > A) in the promoter of the ATG translation initiation site [46,47]. Mutations in the TERT promoter sequence create a new consensus site that binds to the ETS (E-26) transcription factor region, leading to increased protein production resulting in decreased telomerase activity and span [48,49,50]. Another study on HCC patients showed that the TERT promoter in HCC patients was mutated at −297 (C > T) upstream translation site of ATG, transiently spawning the AP2 consensus [51]. Studies have shown that the protein encrypted by RB/E2F genes causes cancer by affecting the TERT promoter activity [52]. The TERT gene is activated when RNA-binding fox-1 homolog 3 (RBFOX3) binds to AP2β, thereby activating telomerase and promoting HCC [53]. SP1 and YAP1 promoted TERT gene expression in the HepG2 cell line, as shown by a previous study [54].

### 4.3. Cell Cycle Regulation

#### 4.3.1. TP53

Liver cancer patients (13–48%) exhibit *TP53* mutations [32,33]. The *TP53* gene quashes the tumors by stunning cancerous cell growth and apoptosis [32]. Mutations in the *TP53* gene of HCC patients were associated with poor health outcomes and prognosis [33]. Non-inflamed tumors exhibit T cell exclusion mediated either through *TP53* gene mutations, also known as an intermediate class, as observed by another study [55]. TP53 mutations, particularly hot spot mutations at R249S and V157F, were associated with poor outcomes and prognosis in HCC patients [56]. 

#### 4.3.2. Tumor Necrosis Factor-Related Apoptosis-Inducing Ligand (TRAIL)/DR4/DR5

The TNF receptor family includes the TRAIL receptor 2/DR5, located on chromosome 8p21-22. Cancer cells were investigated for TRAIL-R2 mutations [57]. A similar study showed that 1% of HCC patients had a mutation in the DR5 domain, suggesting a role in carcinogenesis [57]. TRAIL and IER3 proteins can inhibit Wnt/β-catenin signaling [58]. Research results show that the TRAIL/IER3/β-catenin axis can be a determining element for HCC and can be used as a diagnostic or therapeutic target [58].

TRAIL protein-mediated clustering and oligomerization of DR4/5 recruit multiple adapter factors to form the death-inducing signaling complex (DISC), which then activates caspase-8 and 10 as TRADD and RIP kinases, respectively [59,60,61,62,63].

#### 4.3.3. CDKN2A, CCND1, FGF3, FGF4, or FGF1

CDKN2A mutations have been detected in approximately 8% of HCC patients [36]. CDKN2A is also a tumor suppressor that induces cell cycle arrest in the G1 and G2 phases, making it a potential candidate for HCC treatment. Expression of CDK4/6 and MDM2 is also hindered by this protein, which is considered to be important for oncogenic activity [64]. A recent study showed that loss of CDKN2A in HCC patients reduced the increase in end-stage CDK4/6 inhibitors [65].

In liver cancer, CCND1 and FGF3, FGF4, or FGF19 mutations are found in approximately 5–7% and 4–6% of patients, respectively [36,66]. One study found that the development of CCND1, FGF3, FGF4, or FGF19 was allied with poor prognosis and outcome in individuals who had undergone HCC resection [36]. One study showed that anti-FGF19 combined with antisense RNA-mediated reduction of FGF19 or CCND1 inhibited the 11q13.3 amplicon [67]. 

### 4.4. Oxidative Stress

Hepatocytes contain many fatty acids responsible for oxidative and endoplasmic reticulum (ER) stress. Additionally, these stresses can cause cell damage and inflammation [68]. In animal studies, ER stress has been shown to contribute to NASH-induced HCC due to activation of various pathways, including NF-αB and TNF [69]. A previously published study shows that ER stress mediates increased hepatic steatosis secondary to activation of SREBP1 that initiates adipogenic processes [69]. Endoplasmic reticulum stress combined with steatosis produces ROS in hepatocytes, increasing the magnitude of oxidative stress and oncogenic transformation. These ROS cause lipotoxic apoptosis of hepatocytes and thereby activate macrophages. TNF-α release activates chemokines and growth factors, reducing inflammation in the hepatocyte microenvironment [69]. In addition, ROS generation causes DNA damage owing to mitochondrial malfunction, which contributes to the pathogenesis of HCC in people [70]. 

In prior work, it was shown that mTORC2 activation inside hepatocytes increases the quantity of sphingolipid glucosylceramide and, therefore, ROS production leads to HCC [71]. Impaired cholesterol metabolism contributes to the pathogenesis of HCC [72]. A clinical investigation established the trend of HCC in patients and discovered that NASH posed a larger risk for HCC pathogenesis than NAFLD [73]. 

## 5. Potential Anticancer Mechanism of Nanoquercetin in HCC

Quercetin belongs to the naturally occurring flavonoid class and is widely known for its therapeutic effects, including pro-apoptotic, proliferative, and antioxidant [74]. It is a well-known inhibitor of casein kinase-2α that is responsible for HCC pathogenesis [75]. Some studies have also deciphered the role of casein kinase-2α in the apoptosis mechanism and activation of death receptor pathways [76,77,78]. Moreover, studies showed that nanoformulation of quercetin improves the mechanistic action and therapeutic properties compared to pure quercetin form due to several limitations, including less bioavailability, slow absorption, and short action. Therefore, nanoquercetin showed enhanced anticancer activities by significantly modulating the signaling pathways, as shown below.

### 5.1. Wnt/β-Catenin Signaling Pathway 

The Wnt/β-catenin signaling pathway regulates several biological processes, including cell differentiation, proliferation, migration, and the APC/β-catenin/Tcf pathway [79]. In another in vitro study, it was found that quercetin showed inhibition of SOX2, Nanog, and Oct4 expression along with β-catenin nuclear translocation that, in turn, resulted in downregulated expression of β-catenin-dependent transcriptional activity [80]. In another study, it was found that 20 µM quercetin showed reduced viability through regulating DKK1, 2, and 3 proteins that, in turn, act as checkpoints of Wnt signaling [81].

### 5.2. PI3K/AKT Pathway 

PI3K mediates the translocation process of AKT to the plasma membrane and regulates the mechanism of cell cycle progression, differentiation, cell survival, and cell proliferation [82]. PI3K/AKT is also observed to regulate the expression of Bax (a Bcl-2 protein family member), which is responsible for the anti-apoptotic mechanism [83]. Authors from the previously published study reported the anticancer activity of quercetin against HCC1937 PTEN cancer cell lines through regulation of AKT/PKB phosphorylation [84]. 

In one of the studies, it was suggested that flavonoids directly or indirectly inhibit the mTOR signaling mechanism [85]. It is known that PDK1 is considered to be a major kinase necessary for the development of the mammalian cell. It was found that in the cancer microenvironment, the degree of phosphorylation of AKT kinase at Thr-308 significantly increased [86]. Another study found that quercetin triggers the downregulation of phosphorylation of PDK1 and is hence considered to be the therapeutic target of quercetin and regulatory checkpoints at Ser-473 and Thr-308 [87]. Quercetin is considered to be a broad-spectrum inhibitor of PI3K/AKT1/2, as found in a previous study [88]. Hence, it was considered that quercetin inhibits AKT1/2 by acting directly on inducing Ser/Thr kinase activity and downregulating PI3K. 

### 5.3. JAK/STAT Signaling 

The JAK/STAT signaling mechanism regulates the immune microenvironment, cell death, proliferation, division, and tumor growth. It is known that the JAK/STAT pathway is controlled by ERK, MAPK, and PI3-kinase. It was reported that carcinomas were associated with the deregulation of the JAK/STAT pathway [89]. Qin and coworkers observed the role of quercetin on the JAK/STAT pathway and observed that MGC-803 cells were arrested at the G2/M stage of the cell cycle mediated through p-STAT3 signaling and also reduced the expression of leptin along with its corresponding receptors [90]. It was also reported that quercetin inhibits the IL-6-triggered glioblastoma cell migration, proliferation, and growth by regulating the STAT-3 signaling mechanism mediated through reduced expression of GP130 and JAK1 [91]. 

Quercetin is known to modulate apoptosis through activation of caspase 3, 8, and PARP cleavage that enables the cell to arrest in the sub-G0/G1 phase of the cell cycle along with reduction of p-JAK1, MMP-9, and p-STAT3 expression [92]. Authors from previous studies claimed that quercetin regulates the apoptosis and autophagy mechanism through the expression of caspase-3 and is further inhibited by JAK2 along with cyclin D1 and mTOR, which in turn suppresses STAT3/5 signaling mechanisms [93,94]. Moreover, quercetin was found to show reduced proliferation potential of HCC along with an increased rate of cellular apoptosis due to regulation of the cell cycle through the expression of Cyclin B1 protein [95]. Cyclin B1 is a cell cycle protein that is synthesized in the S and G2/M phases. Therefore, it is believed that quercetin inhibits the cell cycle at the G2/M phase, along with triggering apoptosis. 

### 5.4. MAPK Signaling 

Mitogen-Activated Protein Kinase (MAPK) exhibits three primary classes of kinases, including ERKs, JNK/SAPK, and p38s. It is considered that MAPK 14, 7, and 12 regulate cellular proliferation, gene expression, differentiation, growth, mitosis, and apoptosis [96]. In addition, one study performed on SMMC7221 cells found that quercetin suppresses the proliferation and reduces lipopolysaccharide-initiated oxidation along with the inhibition of the MAPK signaling pathway [97]. It was also found that quercetin significantly suppresses the activity of p38 MAPK in the fibrotic liver of rats [98]. It was found that isoquercetin activates the caspases 3, 8, and 9, which in turn significantly increases apoptosis and triggers the JNK phosphorylation through suppression of ERK and p38 MAPK, as shown in Figure 5 [99]. 

### 5.5. NF-ĸB, p53, and Apoptotic Signaling 

Quercetin triggers the stimulation of 5-fluorouracil-initiated apoptosis in a p53-dependent manner [100]. A similar study also observed that quercetin and p53 work in a synergistic manner [100]. In another study, quercetin, along with doxorubicin, downregulates Bcl-xl in a p53-dependent phase [101]. Quercetin was found to promote cell death-associated gene expression, including p53, along with downregulation of AKT and Bcl-2 expression [102]. It was also found that quercetin suppresses the mTOR expression simultaneous activation of p53 and Sestin-2 via AMPK. In a relevant study performed using nano-quercetin of HepG2 cells, it was observed that it activates the p53-ROS crosstalk and triggers apoptosis along with modification at the epigenetic level and cell cycle arrest at the sub-G phase [103]. Another study showed that quercetin activates the p21, p53, and GADD45 signaling mechanisms along with simultaneous suppression of JNK mediated through Foxo3a [104]. 

## 6. Therapeutic Profile of Other Natural Anticancer Compounds

Exploring other naturally occurring compounds other than quercetin includes resveratrol and curcumin, which have similar functional properties to quercetin for the treatment of hepatocellular carcinoma and to increase the survival outcome in patients undergoing chemotherapy and radiotherapy. Phytochemicals, including resveratrol, curcumin, and quercetin, are widely known for minimizing chemotherapy-associated resistance, and some studies also reported that these phytochemicals enhance the efficacy of ongoing conventional chemotherapy when used as side supplements. 

Resveratrol (3, 5, 4′-trihydroxystilbene) is a stilbenoid polyphenol and phytoalexin that is present in two forms, mainly cis-resveratrol and trans-resveratrol; however, it has been reported that trans-resveratrol is a biologically active form having significant bioactivity and therapeutic characteristics. It is also reported in previously published literature that trans-resveratrol showed 77–80% absorption within the gastrointestinal tract, with around 49–60% removed via the urine [105,106]. Resveratrol, like other phytochemicals, possesses significant antioxidant, anti-inflammatory, anti-proliferative, and anti-angiogenic potential that can serve as a therapeutic target of hepatocellular carcinoma through multiple molecular pathways [107,108]. Hepatocellular carcinoma at various stages, including initiation, promotion, and progression, was significantly hampered by resveratrol, quercetin, and curcumin.

## 7. Protective Mechanism of EVs in HCC

Extracellular vesicles are known to enhance the anticancerous potential by hampering several signaling pathways involved in the metastasis of HCC. In one of the previously published studies, it was found that the Vps4A level is higher in EVs derived from HCC, which inhibits the PI3K-Akt signaling pathway that, in turn, inhibits HCC progression and metastasis [109]. EVs also served as a mediator in the regulation of intracellular micro-RNAs (miRNAs). In one of the studies, it was found that Vps4A exhibits two oncogenic miRNAs (miR-27b-3p and miR-92a-3p), and that is found to be upregulated in SMMC-Vps4A [109]. 

In addition to this, SMMC-Vps4A also possesses miR-193a-3p, miR-320a, and miR-132-3p as tumor suppressor miRNAs [109]. A similar study further detected six tumor suppressor miRNAs (miR-122-5p, miR-33a-5p, miR-34a-5p, miR-193a-3p, miR-16-5p, and miR-29b-3p) that showed an upregulated trend [109]. The findings of this study were supported by the other authors, and they found the role of tumor suppressor miRNAs miR-122-5p, miR-33a-5p, miR-34a-5p, miR-16-5p, and miR-29b-3p in HCC (1010). The authors of a previously published study performed western blotting and found that overexpression of Vps4A leads to the inactivation of the PI3K/Akt signaling pathway that also modulates the miRNAs [109,110]. So, it was concluded that Vps4A showed a therapeutic target against HCC in a miRNA-dependent and independent manner and can be explored as checkpoints for the treatment of patients with HCC. 

Another study found that expression of *SENP3-EIF4A1* and *SENP3-EIF4A1* in secretory EVs suppress HCC proliferation through miR-9-5p mediated action of *ZFP36* [111]. Moreover, IncRNA 85 controls the cancer cell invasion by acting on miR-324-5p by amendable the manifestation of *MMPs*, *ETS1*, and *SP1* in HCC [112]. EVs containing miR-320a showed a protective effect against HCC through suppression of the PBX3/ERK1/2/CDK2 signaling pathway [113].

Mesenchymal stromal cell (MSC)-derived EVs are known to exhibit anticancer properties and can be explored for HCC treatment. Previous research observed that umbilical cord-derived MSCs significantly improved the anti-tumor response of NKT cells in liver cancer by inhibiting oxidative stress [114]. In another study, it was found that miR-122 provides an anticancer effect against HCC by suppressing the PI3-K/Akt signaling pathway, as shown in Figure 6 [115]. In another study, it was found that fibroblast-derived EVs exhibit less quantity of miR-320a and thus inhibit HCC by suppressing the MAPK signaling pathway [113]. Another study claimed that fibroblast-derived EVs were rich in miR-150-3p and exhibited anticancer properties against HCC [116]. Some studies found that the presence of miR-195 in fibroblast-derived EVs suppresses the activation of VEGF, CDC42, CDK1, CDK4, CDK6, and CDC25 and is considered to be a new therapeutic target for HCC [117,118]. In another study, mi331-3p also inhibits the progression of HCC through the regulation of BAK1 [119]. 

## 8. Challenges and Perspectives of Anticancer EVs Biopharmaceuticals 

In the last decades, the unrelenting progress of biologics has encouraged the development of a thorough understanding and technological advancement of biopharmaceutical manufacturing procedures. This sharp evolution has deepened the interest of the biopharmaceutical industries in process analytical technology (PAT), which is known as a system for designing, analyzing, and controlling the manufacturing of products along with ensuring the final quality of the product [120]. Nanoparticles are well known to deliver many biologics, including proteins, peptides, and antibodies. However, such particles face severe challenges, including physiological barriers, fast wash-off from targeted sites, poor permeation–retention effect, etc. With the time spent, technological advancements help researchers overcome several hurdles with the advent of extracellular vesicles. EVs, if compared to synthetic drug delivery nanomaterials, exhibit natural site-targeted features along with improved stability, biocompatibility, and increased bioavailability. Therefore, EVs are considered to be the biggest opportunity for the biopharmaceutical industry to be used as drug/nanoparticle delivery vehicles. Although substantial breakthroughs were fabricated using these engineered EVs as an anticancer therapy, some challenges may still hinder the path to making bench-to-bedside products. The complex structure of EVs is associated with a high degree of inconsistency that might affect the therapeutic properties. Moreover, large-scale isolation and purification approaches of EVs still compromised their yield. Researchers nowadays concentrate their research on testing customized EVs in preclinical animal models, but data is still lacking in clinical trials. Cargo-loading efficiency is still an unaddressed issue and needs serious attention. Summing up, the issues of biosafety, bioavailability, biocompatibility, and biostability are peculiarities for future clinical translational exploration. 

## 9. Conclusions

Quercetin is a polyphenolic flavonoid exhibiting anticancerous features that exert its therapeutic mechanism in hepatocellular carcinoma through dysregulation of several signaling mechanisms, including PI3K/AKT, NF-κB, P53, Wnt/β-catenin, MAPK, JAK/STAT, and the Hedgehog pathway. Moreover, quercetin is known to modulate several intracellular signaling biologics, including TNF-α, Bax, Bcl-2, caspases, and VEGF. The anticancer potential of quercetin was extensively studied in hepatocellular carcinoma. However, the majority of the research was evident in preclinical studies. Studies are lacking in demonstrating clinical trials. EVs derived from mesenchymal lineages were considered to trigger anticancerous effects through several miRNAs and IncRNAs. Not a single study has been conducted in the past that demonstrated the synergistic effect of quercetin and mesenchymal stem cell-derived EVs during the clinical trial phase. There are significantly high expectations of such phase III trials focusing on all stages of HCC.

## Figures and Tables

**Figure 1 cells-13-00638-f001:**
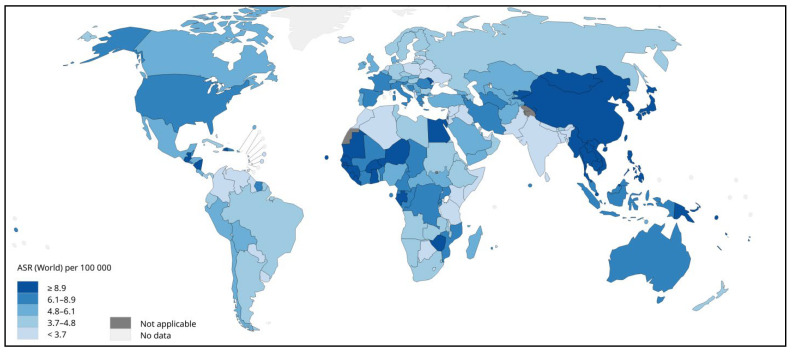
Estimated age-standardized incidence rates (ASR) (worldwide) for liver cancer, both sexes and all ages, in 2020. Adapted from [3].

**Figure 2 cells-13-00638-f002:**
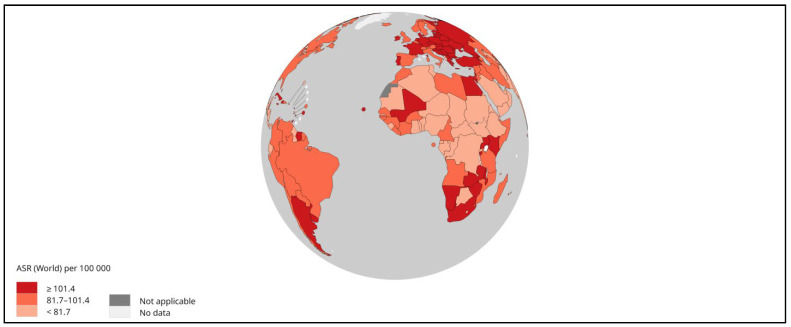
Estimated age-standardized mortality rates (worldwide) in 2020 for liver cancer, both sexes and all ages. Adapted from [3].

**Figure 3 cells-13-00638-f003:**
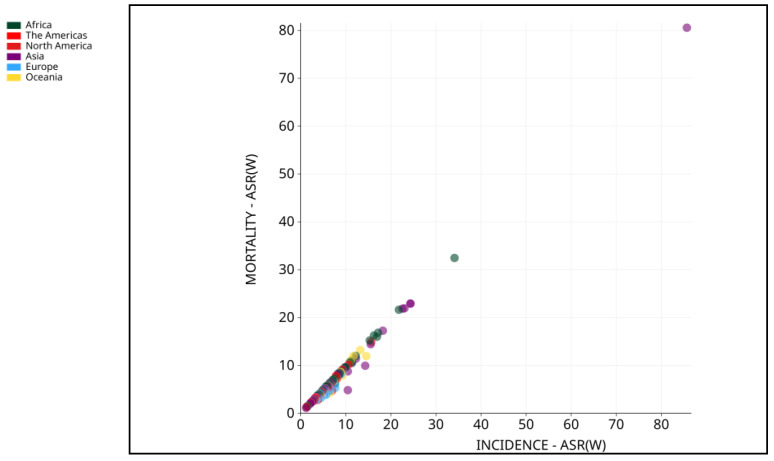
Mortality-ASR (worldwide) vs. incidence-ASR (worldwide) in 2020 for both sexes and all ages. Adapted from [3].

**Figure 4 cells-13-00638-f004:**
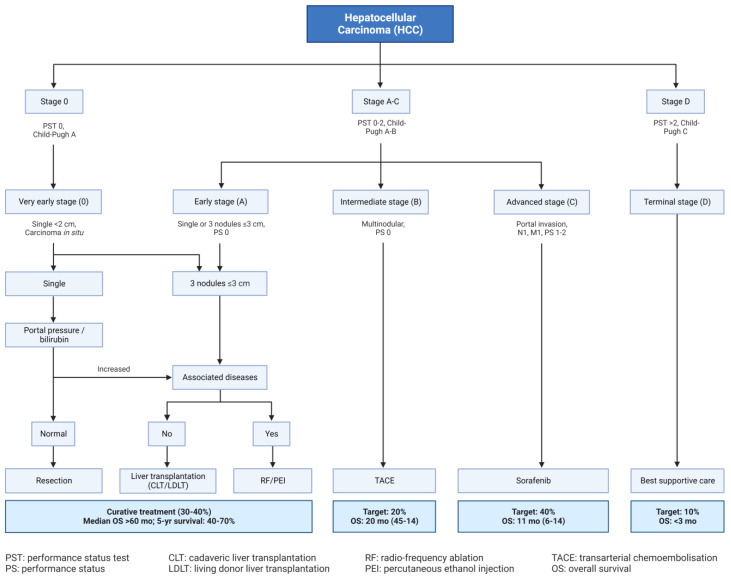
Barcelona Clinic Liver Cancer (BCLC) staging system.

**Figure 5 cells-13-00638-f005:**
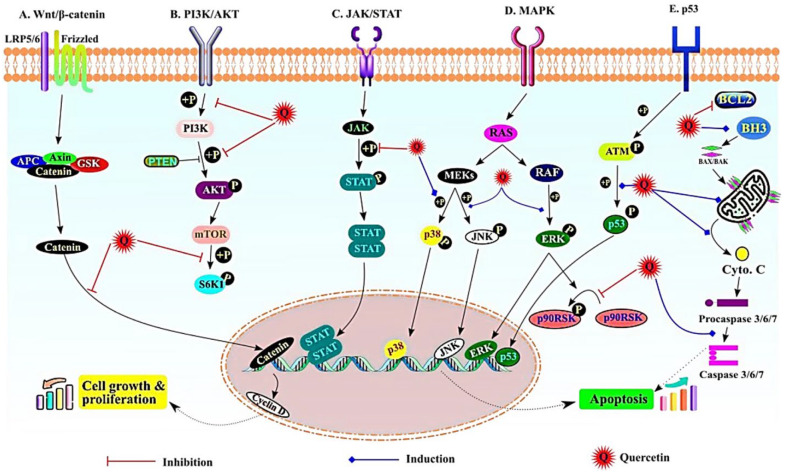
The most important signaling pathways affected by quercetin during cancer prevention. (**A**) Wnt/β catenin pathway—quercetin inhibits β-catenin translocation to the nucleus; (**B**) PI3K/Akt pathway—inhibition of phosphorylation of PI3K, Akt, and S6K; (**C**) JAK/STAT pathway—inhibition of p-STAT formation; (**D**) MAPK pathway—induced phosphorylation of p38, JNK, and ERK; (**E**) p53 pathway—induced phosphorylation of p53 and induction of apoptosis. Adapted from [99] under the terms of the Creative Commons Attribution License (CC BY).

**Figure 6 cells-13-00638-f006:**
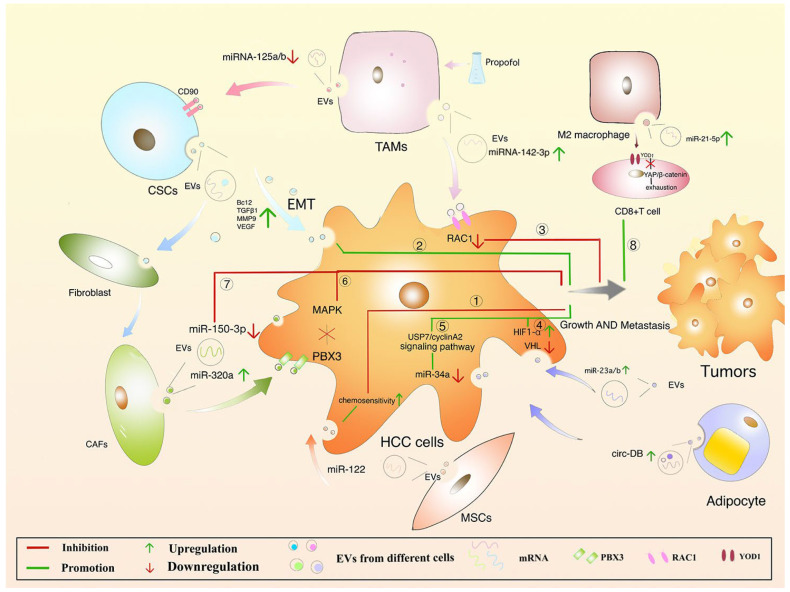
Hepatocellular carcinoma regulations by EVs. Adapted from Ref. [111] under the terms of the Creative Commons Attribution License (CC BY). TAMs, tumor-associated macrophages; CSCs, cancer stem cells; CAFs, cancer-associated fibroblasts; EMT, epithelial–mesenchymal transition.

## Data Availability

No new data were created or analyzed in this study. Data sharing is not applicable to this article.

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
