# Peer review of "Nanoquercetin and Extracellular Vesicles as Potential Anticancer Therapeutics in Hepatocellular Carcinoma"

_cells, 2024, doi:10.3390/cells13070638_

Round 1

Reviewer 1 Report

Comments and Suggestions for Authors

The manuscript provides a comprehensive description of the physiopathology and molecular triggers of hepatocellular carcinoma (HCC). It also offers valuable insights into the mechanisms of action of quercetin in HCC. However, critical information regarding the effective concentration for optimal effects is lacking. Additionally, there is a notable absence of argumentation regarding the necessity of nanometric systems for quercetin delivery. Furthermore, the manuscript lacks details on the nanometric formulation of quercetin, including routes of administration, kinetics, treatment periods, and related considerations. Concerning extracellular vesicles as anticancer vehicles for HCC, the manuscript merely notes the absence of information in the literature, without providing a more in-depth exploration. Notably, there is insufficient justification for conducting a review on this specific topic.

The title, "Synergistic effect of extracellular vesicles and nanoquercetin as anticancer therapeutics in hepatocellular carcinoma," is misleading, as there is no supporting evidence for the indicated synergistic phenomena in the literature.

In alignment with the title of the special topic, it is recommended to broaden the scope by including other natural anticancer compounds such as resveratrol, curcumin, etc., formulated as nanosystems. This would enhance the overall relevance and completeness of the review.

Author Response

Comments 1: [The manuscript provides a comprehensive description of the physiopathology and molecular triggers of hepatocellular carcinoma (HCC). It also offers valuable insights into the mechanisms of action of quercetin in HCC. However, critical information regarding the effective concentration for optimal effects is lacking. Additionally, there is a notable absence of argumentation regarding the necessity of nanometric systems for quercetin delivery. Furthermore, the manuscript lacks details on the nanometric formulation of quercetin, including routes of administration, kinetics, treatment periods, and related considerations. Concerning extracellular vesicles as anticancer vehicles for HCC, the manuscript merely notes the absence of information in the literature, without providing a more in-depth exploration. Notably, there is insufficient justification for conducting a review on this specific topic.]

Response 1: Thank you for pointing this out. I/We agree with this comment. It is to bring into your kind notice no such studies were found in previously published literature those quoted optimal concentration of nanoquercetin in treatment of HCC, so such studies were not included in this article. As it need more laboratory based research to conclude this statement. Regarding the nanometric delivery system including routes of administration, kinetics, treatment periods, and related considerations it is still not clear about the comment that which nanometric system reviewer is concerned about, if the reviewer will share any published literature, it will be a help for us to provide precise information about reviewer concern.

Comments 2: [The title, "Synergistic effect of extracellular vesicles and nanoquercetin as anticancer therapeutics in hepatocellular carcinoma," is misleading, as there is no supporting evidence for the indicated synergistic phenomena in the literature.]

Response 2: Agree. I/We have, accordingly, revised the title of the manuscript.

Comments 3: [In alignment with the title of the special topic, it is recommended to broaden the scope by including other natural anticancer compounds such as resveratrol, curcumin, etc., formulated as nanosystems. This would enhance the overall relevance and completeness of the review.]

Response 3: Agree. I/We have, accordingly, added the paragraph focusing on resveratrol and curcumin.   

Reviewer 2 Report

Comments and Suggestions for Authors

In the present review, the authors discussed the synergistic effect of EVs and nano quercetin against HCC. The manuscript is well written, to the point, of significant interest in the field. However, a few minor modifications may improve the manuscript. My comments are as follows:

1. Abstract: line 15: please remove 'to vanish' as it is difficult to achieve 100% efficiency. 

2. The authors discussed the beneficial effects of EVs and nano quercetin in synergy. It would be good if the authors also discuss the potential effects of quercetin loaded EVs in a brief paragraph. Would it be more effective?

3. A paragraph should be included discussing the target specificity of EVs and nano quercetin against HCC.

4. Kindly discuss very briefly how the formulation could be effective against other form of cancers. 

5. Briefly discuss how EVs-based drug delivery is better than other conventional approaches.

6. Does alcohol consumption itself induce EVs generation from HCC? What is the effect of these EVs if any? Please discuss.

7. It would be nice to include a table describing the checkpoint targets of HCC, including the pathway, target genes, mechanism of action, effective drugs, and references. 

8. Page 10: line 374-375: Include the reference 'PMID: 31341019', where the role of miR221 in modulating PI3K/AKT pathway to induce cancer progression is discussed.

9. Please elaborate the abbreviations used in the text, many remains unabbreviated. 

Author Response

Comments 1: [1. Abstract: line 15: please remove 'to vanish' as it is difficult to achieve 100% efficiency.]

Response 1: Thank you for pointing this out. I/We agree with this comment. Therefore, I/we have removed the word “vanish”.

Comments 2: [The authors discussed the beneficial effects of EVs and nano quercetin in synergy. It would be good if the authors also discuss the potential effects of quercetin loaded EVs in a brief paragraph. Would it be more effective?]

Response 2: Thanks for your suggestion. But it will be controversial because quercetin size is more than the size of the EVs so practically and theoretically it will not be possible to incorporate the quercetin as such into the EVs without converting them into nanoparticle. So, we have not added such paragraph.

Comments 3: [A paragraph should be included discussing the target specificity of EVs and nano quercetin against HCC.]

Response 3: Agree. I/We have, added the information already in the submitted manuscript.

Comments 4: [Kindly discuss very briefly how the formulation could be effective against other form of cancers.]

Response 4: This manuscript is dedicated to the hepatocellular carcinoma so not included other cancer otherwise it will be deviation from our objectives and it will be very vast topic to discuss all forms of cancers.

Comments 5: [Briefly discuss how EVs-based drug delivery is better than other conventional approaches.]

Response 5: Agree. Please clear the things about the conventional approaches that reviewer is talking about so that precisely we can address. But the present manuscript is sufficient to justify the EVs and nanoquercetin targets in HCC.

Comments 6: [Does alcohol consumption itself induce EVs generation from HCC? What is the effect of these EVs if any? Please discuss.]

Response 6: This is outside the scope of this manuscript because there are various factors that modulate the EVs generation.

Comments 7: [It would be nice to include a table describing the checkpoint targets of HCC, including the pathway, target genes, mechanism of action, effective drugs, and references.

Response 7: It is included into the text itself.

Comments 8: [Page 10: line 374-375: Include the reference 'PMID: 31341019', where the role of miR221 in modulating PI3K/AKT pathway to induce cancer progression is discussed.]

Response 8: miR221 is not discussed in the present manuscript.